# Viral Interference between the Insect-Specific Virus Brejeira and the Saint Louis Encephalitis Virus In Vitro

**DOI:** 10.3390/v16020210

**Published:** 2024-01-30

**Authors:** Ana Cláudia Ribeiro, Lívia Martins, Heloisa Silva, Maria Nazaré Freitas, Maissa Santos, Ercília Gonçalves, Alana Sousa, Ivy Prazeres, Alessandra Santos, Ana Cecilia Cruz, Sandro Silva, Jannifer Chiang, Livia Casseb, Valéria Carvalho

**Affiliations:** 1Post-Graduation Program in Virology, Evandro Chagas Institute, Ananindeua 67030-000, Pará, Brazil; liviamartins@iec.gov.br (L.M.); maria_freitas17@hotmail.com (M.N.F.); anacecilia@iec.gov.br (A.C.C.); janniferchiang@iec.gov.br (J.C.); liviacasseb@iec.gov.br (L.C.); 2Department of Arbovirology and Hemorrhagic Fevers, Evandro Chagas Institute, Ananindeua 67030-000, Pará, Brazil; heloiza.teixeira1997@gmail.com (H.S.); maissasantos@iec.gov.br (M.S.); erciliagoncalves@iec.gov.br (E.G.); alanawatanabe@yahoo.com.br (A.S.); ivyprazeres@iec.gov.br (I.P.); alebio2005@yahoo.com.br (A.S.); spatroca@gmail.com (S.S.)

**Keywords:** arbovirus, Saint Louis encephalitis virus, insect-specific viruses, Brejeira virus

## Abstract

The Saint Louis encephalitis virus (SLEV) is an encephalitogenic arbovirus (*Flaviviridae* family) that has a wide geographical distribution in the western hemisphere, especially in the Americas. The negevirus Brejeira (BREV) was isolated for the first time in Brazil in 2005. This study aimed to verify the existence of a possible interfering effect of BREV on the course of SLEV infection and vice versa. We used clone C6/36 cells. Three combinations of MOIs were used (SLEV 0.1 × BREV 1; SLEV 1 × BREV 0.1; SLEV 1 × BREV 1) in the kinetics of up to 7 days and then the techniques of indirect immunofluorescence (IFA), a plaque assay on Vero cells, and RT-PCR were performed. Our results showed that the cytopathic effect (CPE) caused by BREV was more pronounced than the CPE caused by SLEV. Results of IFA, the plaque assay, and RT-PCR showed the suppression of SLEV replication in the co-infection condition in all the MOI combinations used. The SLEV suppression was dose-dependent. Therefore, the ISV Brejeira can suppress SLEV replication in *Aedes albopictus* cells, but SLEV does not negatively interfere with BREV replication.

## 1. Introduction

Arboviruses are viruses transmitted by hematophagous insects [1]. These viruses can cause diseases, and many are considered problems of public health, such as the Saint Louis encephalitis virus (SLEV) (*Orthoflavivirus louisense*), which belongs to the *Flaviviridae* family and the *Orthoflavivirus* genus. SLEV has a positive-sense single-stranded RNA genome and belongs to the Japanese encephalitis group. The viral particle is spherical and possesses an icosahedral nucleocapsid (40 nm in diameter) [1,2,3,4,5].

The first isolation of SLEV was obtained from the brain suspension of a case of acute encephalitis during an urban outbreak of the disease in the city of Saint Louis, United States, in 1933 [6,7]. In Brazil, the first isolate was obtained in 1960 from a pool of *Sabethes belisarioi* mosquitoes captured on the Belém–Brasília highway, and it has subsequently been detected in arthropods by virus isolation and serological surveys in wild birds and mammals, non-human primates, marsupials, sloths, armadillos, and horses [8,9,10,11].

SLEV has a wide geographical distribution in the west, including the United States, Canada, and Central and South America [4,6]. In Brazil, it has been reported circulating in Pará, São Paulo, and Mato Grosso, among others. The prevalence of hemagglutination-inhibiting (HI) antibodies to the virus in the Brazilian Amazon population ranges from 1 to 5% [1,9].

The SLEV maintenance cycle involves mosquitoes, wild birds, equines, and humans. Mosquitoes of the *Culex* genus, especially *Culex declarator* and *Culex coronator*, are the main SLEV vectors. In North America, *Culex pipiens*, *Culex quinquefasciatus*, and *Culex tarsalis* mosquitoes are also important vectors. Wild birds serve as primary amplification hosts for the virus, while equines, such as horses, do not show high viremia or clinical disease; still, antibodies are detected in these infected animals. Humans are considered accidental and terminal hosts because, once infected, they have low viremia, which is insufficient to infect mosquitoes [1,4,8,10,11,12,13].

The infection caused by SLEV in humans can be asymptomatic and symptomatic, ranging from mild febrile illness, with fever, headache, nausea, vomiting, and tiredness, to fatal severe neuroinvasive disease, aseptic meningitis, or encephalitis. Encephalitis is one of the most severe clinical manifestations of arbovirus infections and can lead to death or severely disabling effects [8,14].

In the last few decades, molecular techniques have evolved, increasing the number of new viruses identified, including insect-specific viruses (ISVs) [15]. ISVs naturally infect mosquitoes and replicate in mosquito cells in vitro. These viruses are distributed in several viral families, including arboviruses such as the *Flaviviridae* and *Togaviridae*. In addition, other ISVs belong to the Negevirus taxon, including the Brejeira virus (BREV) [16,17]. According to the classification of negeviruses proposed by Kallies and colleagues (2014) [18], BREV is included in the genus *Nelorpivirus* [18,19].

BREV is a positive-sense, single-stranded RNA with three well-defined open reading frames (ORF) (ORF-1, ORF-2, and ORF-3) measuring 7014 nt, 1203 nt, and 624 nt, respectively. ORF-1 (large) encodes the viral polymerase, while ORF-2 (medium) encodes the glycoproteins and ORF-3 (small) encodes the membrane proteins [19].

This virus was first isolated in Brazil in 2005, in the northern region, in Pará; it was also isolated in Colombia in Córdoba in 2013 and Mato Grosso do Sul (Pantanal, Brazil) in 2010. In most reports, the virus has been isolated from hematophagous arthropods of the genus *Culex* sp. [19,20,21]. In the last ten years, six strains of BREV were isolated from pools of *Culex (Cux.)* species collected in the regions of Canaã de Carajás and Curionópolis, Pará State, Brazil, in 2014 and 2015, respectively [22].

Some studies have shown the interference of ISVs in the replication of arboviruses, mainly those ISVs from the *Flaviviridae* and *Togaviridae* families. Because of this, ISVs have been seen as potential biological control tools for arboviruses and platforms for the development of vaccines and laboratory diagnostics. New tools to fight against arboviruses are essential, given that these viruses can cause outbreaks or epidemics; there are no specific drugs for the treatment of patients, and vaccines are scarce [15].

Some published studies have shown the potential of ISVs to interfere with arbovirus replication. For example, *Ae. aegypti* mosquitoes were previously infected with ISVs, namely cell fusion agent virus (CFAV) co-infected with the arboviruses Dengue virus (DENV) (*Orthoflavivirus denguei*) and Zika virus (ZIKV) (*Orthoflavivirus zikaense*), and both arboviruses in the study showed a reduction in their titers [23]. Another ISV that was investigated was the Nhumirim virus (NHUV), which caused a 40% reduction in the transmissibility of West Nile virus (WNV) (*Orthoflavivirus nilense*) in colonies of *Ae. albopictus* mosquitoes infected with NHUV [24].

Another application of ISVs is in the development of vaccine platforms for arboviruses. By using reverse genetics techniques, ISVs can be used as backbones for the insertion of arbovirus proteins, resulting in the formation of chimeric viruses, which are unable to replicate in vertebrate hosts but induce neutralizing antibodies and cellular responses, thus resulting in a promising strategy to prevent the emergence or reduce the morbidity and mortality caused by arboviruses [25,26,27]. In addition, the chimeras produced can also be used as a safer viral antigen for laboratory tests, such as the enzyme-linked immunosorbent assay (ELISA) [26,27].

Although there is already some information on the interference caused by ISVs, very little research involving negeviruses and arboviruses has been published. One of these studies was published in 2020 by Patterson and colleagues [28], who demonstrated that the Negev virus (NEGV) can inhibit the replication of the arboviruses belonging to the *Alphavirus* genus, Chikungunya virus (CHIKV), and Eastern equine encephalitis virus (EEEV) in co-infected mosquito cells. Our study assessed whether the BREV negevirus interferes with SLEV replication when both viruses co-infect mosquito cells in vitro and whether the reverse also exists, i.e., whether the SLEV virus interferes with BREV replication.

## 2. Materials and Methods

### 2.1. Viral Samples

The viral samples used in this study were part of the arbovirus collection of the Arbovirology and Hemorrhagic Fevers Department of the Evandro Chagas Institute (SAARB-IEC), Ananindeua, Pará, Brazil, a national benchmark for the diagnosis of arboviruses. The SLEV viral isolate (BE AR 23379) came from *Sabethes belisarioi* mosquitoes, captured in 1960 on the Belém–Brasília highway, Brazil, and BREV (BE AR 805511) came from *Culex (Cx.)* mosquitoes from the Canaã dos Carajás region, Pará State, Brazil, in 2014.

### 2.2. Cells

Clone C6/36 cells (ATCC: CRL 1660) from *Aedes albopictus* mosquitoes [29] and Vero cells from African green monkeys *Chlorocebus sabaeus* (ATCC: CCL-81) were used. The C6/36 cells were kept in an incubator at 28 °C and supplemented with medium L-15 (Gibco, Waltham, MA, USA) with L-glutamine (Gibco, Waltham, MA, USA), supplemented with 5% fetal bovine serum (FBS) (Gibco, Waltham, USA), tryptose phosphate (Himedia, Mumbai, India) (2.95%), antibiotics (penicillin 10,000 U/L and streptomycin 10,000 g/L) (Gibco, Waltham, MA, USA), and non-essential amino acids (10 mL/L) (Baktron Microbiology, Rio de Janeiro, Brazil). In contrast, the Vero cells were kept in an incubator at 37 °C in medium 199 (Gibco, Waltham, MA, USA) supplemented with 5% FBS (Gibco, Waltham, MA, USA) and 1% antibiotics penicillin/streptomycin (Gibco, Waltham, MA, USA). These cells were maintained in the laboratory by weekly cell passage using the protocols of Barbosa ML, Rocco I M, Felippe JMMS, Cruz AS (1993) [30] and Ammerman NC, Beier-Sexton M, Azad AF (2008) [31].

### 2.3. Preparation and Titration of the Viral Stock

The viral stock for the viruses under study was produced in C6/36 cells by inoculating 100 µL of the virus sample into 25 cm^2^ cell bottles. Inoculation was carried out using the viral adsorption method. They were then observed daily under an inverted optical microscope, and when they showed a cytopathic effect, they were frozen at −80 °C in a freezer. After this, aliquots of 100 µL of this material were created in Eppendorf tubes to be stored in a freezer at −80 °C. Subsequently, the viral titer of SLEV was obtained by plaque assay, and the titer of BREV was obtained by TCID_50_ [32,33]. To use the titer values (PFU/mL) in the MOI formula, we transformed the BREV titer from TCID_50_/_mL_ to PFU/mL using the formula TCID_50_/mL titer × 0.7 = titer in PFU/mL [34].

### 2.4. Tissue Culture Infectious Dose 50% (TCID_50_)

Vero cells were seeded in 24-well plates. When the cells were confluent, we prepared 10× serial dilutions of the virus. After removing the media, we inoculated 100 µL of each sample into four wells each. The plates were incubated at 37 °C for one hour, carefully rocking them every 15 min to allow better virus distribution on the cell monolayer. After incubation, we added 0.5 mL of 199 maintenance media with 2% FBS to each well. The plates were further incubated at 37 °C, and cells were monitored for CPE for seven days.

On day seven post-inoculation, the plates were fixed using cold 25% methanol for 10 min. The methanol was removed, and the cells were stained with 0.5% crystal violet and incubated at room temperature for 30 min. The plates were rinsed in water and left to dry. The CPE was graded in each well, and the TCID_50_ was calculated using the Reed–Muench method (1938) [33].

### 2.5. Co-Infection Experiments in Cell Culture between SLEV and BREV

C6/36 clone cells were used for co-infection since ISVs do not replicate in vertebrate cells. The following combinations of infection multiplicities (MOIs) were used: MOI 1 for BREV and MOI of 0.1 for SLEV; MOI 0.1 for BREV and MOI of 1 for SLEV; MOI 1 for BREV and MOI of 1 for SLEV. First, C6/36 cells were seeded in 12-well plates. Before infection, the volume of each virus (stock) was calculated according to the number of cells per well. The growth medium (5% FBS) of the plates was discarded, and the cells were then inoculated in triplicate for each of the following conditions: (1) concurrent infection with BREV and SLEV; (2) infection with BREV only (positive control); (3) infection with SLEV only (positive control); (4) negative control (uninfected cells). Subsequently, the cells were incubated at 28 °C for one hour, rocking the plates gently every 15 min. After this, maintenance medium (2% FBS) was added to the plates and incubated for seven days at 28 °C. Then, the cells were observed daily under an inverted optical microscope for possible CPE monitoring of the cell monolayer. The cells were collected on slides for indirect immunofluorescence (IFA) at the following time points: immediately after inoculation and 06 h, 24 h, 48 h, 72 h, 96 h, 120 h, 144 h, and 168 h post-infection. Cell suspensions (cells + supernatant) were collected at each time point of the viral kinetics.

### 2.6. Indirect Immunofluorescence (IFA) and Viral Titration

An indirect immunofluorescence (IFA) assay was performed to evaluate the decrease or increase in positivity (fluorescence) for SLEV in the co-infected cells compared to the SLEV-positive control cells. The slides with cells obtained from the kinetics described in Section 2.5 were fixed in acetone P.A. The reaction was then performed according to the adapted protocol of Gubler and colleagues (1984) [35] using polyclonal anti-SLEV antibodies (produced in-house). The images were acquired on a BX51 microscope using a UPlanFL N 20×/0.5 camera, a WB objective lens, and U-25ND filters (Olympus, Tokyo, Japan).

In addition, the cell suspensions collected in the kinetics were titrated by plaque assay on Vero cells using a protocol adapted from Malewicz and Jenkin, 1979 [32]. Titration was performed to compare the SLEV titer in the SLEV-positive control and co-infection conditions.

### 2.7. Real-Time Reverse Transcription–Polymerase Chain Reaction (Real-Time RT-PCR) for SLEV and BREV

RNA was extracted from the cell supernatant using the Maxwell 16 equipment with the Maxwell 16 Total RNA purification kit (Promega, Madison, WI, USA), following the manufacturer’s instructions.

The reverse transcription technique followed by PCR was used to obtain the number of viral copies. We used the Superscript III Platinum One-Step qRT-PCR System commercial kit with Rox separated, according to the manufacturer’s instructions (Invitrogen, Waltham, MA, USA). The RT-PCRs were performed on the ABI PRISM 7500 standard (Applied Biosystems, Waltham, MA, USA).

The reactions of all the viruses investigated comprised a final reaction of 25 μL, containing 12.5 μL of 2× reaction mix with Rox, 5.5 μL of nuclease-free water, 1.0 μL of SLEV/BREV F/R primer mix, 0.5 μL of SLEV/BREV probe, 0.5 μL of SIII/TaqPI enzyme mix, and 5.0 μL of total RNA, using the following cycling conditions: 1 × 45 °C cycle for one h and 94 °C for 3 min; 40 × 94 °C cycles for 30 s, 55 °C for one min; and 68 °C for 3 min. The respective primers and probes highlighted in Table 1 were used.

### 2.8. Statistical Analysis

The analysis of statistical significance was performed in Prism version 9 using the variance analysis with the two-way ANOVA criterion and Bonferroni’s multiple comparisons test, and the R program was used to construct the charts using the ggplot2 library while adopting a significance level of 95.00% (*p* ≤ 0.0001).

## 3. Results

### 3.1. Viral Titers

The titer of the SLEV viral stock was 2.1 × 10^6^ PFU/mL, and for BREV, it was 0.7 × 10^7^ PFU/mL.

### 3.2. SLEV Co-Infection and BREV in C6/36 Cells

#### 3.2.1. Observing the Cytopathic Effect (CPE)

C6/36 cells were inoculated simultaneously with BREV and SLEV. When BREV was ten times more concentrated than SLEV, the appearance of CPE was observed, starting six hours after infection, both in the co-infection and in the BREV-positive control, with characteristics such as suspended cell clumps, evolving into elliptical cells with a reduction in size, dead cells, and spacing of the cell monolayer. In contrast, the SLEV-positive control began to show CPE starting 24 h after infection, with clumps of cells adhered to the cell layer, dead cells, and cell monolayer destruction. In the co-infection, the prevalence of the characteristics of the BREV CPE compared to the SLEV CPE could be observed (Figure 1).

When the MOI concentrations were inverted, with Brejeira negevirus being ten times less concentrated than SLEV, CPE occurred in the co-infection starting at 24 hpi and also in the BREV-positive control. Even at a lower concentration, more evident characteristics of the BREV CPE than the SLEV CPE (ten times higher) could be observed, especially with the formation of suspended cell clumps. The SLEV-positive control showed CPE at 24 hpi, with the formation of clumps and spacing in the monolayer evolving to the disintegration of the cell monolayer throughout the viral kinetics (Figure 1).

On the other hand, when the concentrations were equalized in the same proportion (1.0) for BREV and SLEV, CPE was observed from 24 hpi, as was found in the BREV- and SLEV-positive controls, with the formation of cell clumps, spacing, and evolving to the destruction of the cell monolayer (Figure 1).

Thus, BREV, whether at lower, higher, or equivalent MOI concentrations, showed a more evident CPE than the SLEV in *Aedes albopictus* cells (C6/36).

#### 3.2.2. Detection of SLEV Viral Antigens by IFA

The indirect immunofluorescence serological test was performed to detect SLEV. It was observed that the co-infected cells in all MOI combinations were negative for this virus throughout the kinetics, except when the concentration of SLEV was ten times higher than that of BREV, when negativity was observed up to 96 hpi, with the appearance of some positive cells from 120 hpi, continuing until 168 hpi (less than 25% of positive cells). The SLEV-positive controls showed the increasing detection of positive fluorescence starting at 24 hpi (Figure 2).

#### 3.2.3. SLEV Titer Curve

Through quantitative analysis by plaque assay on Vero cells, the progressive growth of the SLEV titer during viral kinetics (Figure 3A–C) could be observed in all three MOI combinations tested.

The results show that in the co-infection in which SLEV was ten times less concentrated than BREV, it was possible to detect SLEV from the first few hours of infection, reaching the maximum titer in the first 24 h (4.6 × 10^4^ PFU/mL), with progressive regression after this until no infectious plaques formed at the last time point (168 hpi). Compared to the SLEV-positive control titers at the 144 hpi time point (3.3 × 10^7^ PFU/mL), there was a 100,000-fold reduction in SLEV viral replication when co-infected with the ten-fold more concentrated BREV (4.1 × 10^2^ PFU/mL), with a *p*-value < 0.0001, which is extremely significant. It could also be observed that at the last time point of 168 hpi (viral titer of the positive control equal to 2.3 × 10^6^ PFU/mL), there was a 1,000,000-fold reduction in the replication of SLEV co-infected with BREV. It can be seen that when the negevirus is more concentrated in vitro, there is a reduction in SLEV replication (Figure 3A).

When the concentrations were reversed, when SLEV was ten times more concentrated than the Brejeira negevirus, under this condition, it was possible to detect viral titers for SLEV in the first few hours of infection, when, at time point 0 hpi, a titer of 3.8 × 10^4^ PFU/mL was obtained, with the viral titer decreasing to 1.6 × 10^2^ PFU/mL at the last time point of 168 hpi. Compared with the SLEV-positive control’s titer, the maximum viral titer peak reached was at the 120 hpi time point (1.8 × 10^7^ PFU/mL); it was at this point that the greatest reduction (10,000 times) in the replication of SLEV co-infected with ten times less concentrated BREV occurred, showing extremely significant *p*-values (*p* < 0.0001). Thus, it can be seen that there was also a reduction in the replication of SLEV, even when it was placed in a higher concentration than the negevirus (Figure 3B).

When we equated the concentrations of BREV and SLEV in the co-infection, as in the previous conditions, it was possible to detect viral titers for SLEV from the first few hours of infection, regressing to 4.8 × 10^2^ PFU/mL at the last time point analyzed (168 hpi). Compared with the SLEV-positive control (at 120 hpi: 5 × 10^6^ PFU/mL), we found a 1000-fold reduction in the titer of SLEV co-infected with BREV (1.7 × 10^3^ PFU/mL), with extremely significant *p*-values (*p* < 0.0001). Of all the conditions investigated, this was the one that showed the smallest reduction in SLEV replication, although a decrease in SLEV titers was still observed (Figure 3C).

#### 3.2.4. Viral Genome Detection for SLEV and BREV

The Ct values for the arbovirus SLEV and the negevirus BREV could be obtained; it is important to emphasize that the number of Ct is inversely proportional to the viral titer.

All positive controls (SLEV or BREV) showed Ct values decreasing during the viral kinetics, which means that the viral load increased during the kinetics (Figure 4A,B). In all the experiments, the negative controls tested for the viruses under study were included in each hpi.

Regarding the detection of the SLEV genome, in the co-infection condition where the SLEV concentration was ten times lower than the BREV, it was observed that the SLEV Ct increased progressively, which corresponded to a decrease in the SLEV titer (*p*-values between 0 hpi and 168 hpi varying from *p* = 0.0175 to <0.0001). When the concentration of SLEV was ten times higher than that of BREV, there was an oscillating increase in Ct up to 96 hpi (decrease in viral titer) (*p* < 0.0001); however, a consecutive increase and decrease were observed, showing that when SLEV is more concentrated, it has a more competitive profile. When the viral concentrations were equalized, there was a decrease in the SLEV titer in the co-infection compared to the SLEV-positive control; however, the Ct of the co-infection, although oscillating, decreased over time with a *p*-value < 0.0001, extremely significant at 168 hpi (Figure 4A).

The BREV genome detection analyses showed that, when looking at the curve as a whole, SLEV did not interfere with BREV replication. At some points during the kinetics, there was an increase in the Ct (decrease in titer) of BREV in the co-infection, which was observed mainly when the MOIs of the two viruses were equalized, showing a more competitive profile of SLEV. In this last condition, the *p*-values showed a variation from extremely significant, with *p* < 0.0001 (0 hpi), *p* = 0.0031 (24 hpi), and *p* = 0.0002 (48 hpi), to very significant, with *p* = 0.0012 (72 hpi) (Figure 4B).

## 4. Discussion

Negeviruses are organisms that have not yet been well explored by science and which, due to the evolution of genomic sequencing techniques and entomovirological surveillance, are being increasingly detected and discovered. It was within this framework that it was possible to isolate and sequence the viral strains used in the herein study, such as the BREV strain, which came from epidemiological surveillance actions in areas affected by anthropogenic activity in the southeastern region of the state of Pará, Brazil, as well as the SLEV strain, which was isolated from mosquitoes in the area of the Belém–Brasília highway (km 94), through arbovirus surveillance actions.

Several studies on ISVs, including negeviruses, have shown that these viruses only replicate in the cells of hematophagous arthropods and do not replicate in vertebrate cells. The C6/36 cells from *Aedes albopictus* used in our study proved to be very suitable for the observation of the evolution of the infection of negeviruses, such as BREV, which corroborates the work of Vasilakis and colleagues (2013) [16], who carried out experiments with six ISVs, NEGV, Piura virus (PIUV), Dezidougou virus (DEZV), Ngewotan virus (NWTV), Loreto virus (LORV), and Santana virus (SANV), which were inoculated into C6/36, Vero, and BHK-21 cell lines and also into newborn mice (intracerebral inoculation). The viruses caused CPE only in C6/36 cells, and no inoculated mice became ill, which shows the restriction of ISV replication to only mosquito cells, a fact that, according to the authors, can be explained by the defective RNA interference response that occurs in the C6/36 cell [16,36,37].

BREV showed a well-defined CPE with the formation of suspended clumps, elliptical cells, and loose dead cells, which had already been reported by Ribeiro and colleagues [21]. Our experiments showed that the characteristic CPE of BREV stood out over the CPE of SLEV, which demonstrated the interference caused by BREV in SLEV replication, regardless of the concentrations of MOIs used. On the other hand, the replication of the negevirus showed no interference when placed in the presence of SLEV in the same condition (Figure 1).

Regarding the detection of the SLEV antigen in infected cells (Figure 2), this study showed that BREV suppressed SLEV replication in all three conditions under study, namely MOI 1.0 for BREV and MOI 0.1 for SLEV; MOI 0.1 for BREV and MOI 1.0 for SLEV; and MOI 1.0 for both BREV and SLEV.

Concerning the evaluation of SLEV titers (PFU/mL) (Figure 3A), it was possible to see that the arbovirus under study showed progressive growth until the end of the viral kinetics, unlike what was observed in the condition of co-infection with BREV, in which a reduction in the SLEV titer was observed in all the MOI combinations used. Thus, it was evident that the ISV BREV inhibited SLEV replication in C6/36 cells in the MOI where BREV was ten times more concentrated than SLEV, with a 100,000-fold decrease in the SLEV titer, even reaching a 1,000,000-fold reduction in the replication of SLEV co-infected with BREV at the last time point (168 hpi). Meanwhile, when the concentration of this negevirus was ten times lower (Figure 3B), inhibition occurred at 120 hpi, with a 10,000-fold reduction in the titer of SLEV infected with BREV (*p* < 0.0001). Finally, when the two viruses were matched at equal concentrations (Figure 3C), inhibition occurred throughout the viral kinetics, reaching, at 120 hpi, a 1000-fold reduction in the titer of SLEV co-infected with BREV, with statistically significant values (*p* < 0.0001).

When the co-infections and the controls using RT-qPCR for SLEV were analyzed (Figure 4A), the control kept increasing throughout the viral kinetics up to 168 hpi; when compared with the co-infection with BREV, the reduction in the cycle threshold (Ct) was remarkable, with an extremely significant statistical difference from 6 hpi onwards (*p* < 0.0001), even when BREV was ten times less concentrated, thus suggesting that in all three conditions studied, BREV inhibited SLEV.

Furthermore, the findings herein show that BREV’s interference with SLEV is dose-dependent, whereby, if the dose of SLEV is much higher than that of the negevirus in question, replication inhibition may not occur.

By analyzing the competitive profile between the two viruses studied, BREV and SLEV, in C6/36 cells, BREV showed the greatest competitive advantage over the encephalitogenic arbovirus (SLEV), which was evident when observing that the BREV CPE was predominant in comparison to the arbovirus under study, and the RT-qPCR results for BREV (Figure 4B) showed that there was possibly no significant difference in the Ct value (at almost all the kinetic time points) between the BREV-positive control and the condition in which SLEV was also present, showing that SLEV did not interfere with BREV replication in *Aedes albopictus* mosquito cells (Figure 4B).

Several studies have shown the ability of ISVs to suppress the replication of arboviruses, especially those of the viral genera *Orthoflavivirus* and *Alphavirus*. For example, Romo and colleagues (2018) [38] studied the ability of an insect-specific flavivirus, Nhumirim virus (NHUV), to interfere with the replication of WNV and SLEV flaviviruses. The authors observed a reduction in WNV, Japanese encephalitis virus (JEV) (*Orthoflavivirus japonicum*), and SLEV titers in C6/36 cells. A study by Schultz and colleagues (2018) [39] demonstrated that the dual infection of ISVs Phasi Charoen-like virus (PCLV) and CFAV in Aa23 cell culture (*Aedes albopictus*) negatively impacted ZIKV replication using both a low MOI of 0.1 and a high MOI of 10; moreover, when using this same double-infected strain, the same reduction of one log (90%) in DENV serotype 2 was observed.

Previous studies have also evaluated the antiviral potential of other insect-specific flaviviruses, such as the Parramatta River virus (VPaR), which has been shown to inhibit the replication of DENV serotype 3 and WNV in C6/36 clone cells [40], in addition to the Palm Creek ISV (PCV) (*Flaviviridae* family), which also suppressed replication in the C6/36 cells of two arboviruses, WNV and Murray Valley encephalitis virus (MVEV) [41].

Most published studies on the interference between ISVs and arboviruses involve insect-specific flaviviruses and arboviruses from the same viral genus (*Orthoflavivirus*). However, a diverse range of ISVs belonging to other viral families and genera deserve to be studied, since viruses from different genera can often infect the same mosquito vector and interact. In this context, Patterson and colleagues (2020) [28] published the first study on the interaction between negeviruses and arboviruses of the *Alphavirus* genus. In their study, the researchers observed that the negevirus, Negev virus (NEGV), reduced the replication of Venezuelan equine encephalitis virus (VEEV) and CHIKV during co-infections in mosquito cells.

Our study is one of the first to seek a better understanding of the interaction between BREV, which is a negevirus, and an arbovirus of public health importance. SLEV has emergency potential in Brazil and worldwide and is responsible for causing encephalitogenic diseases in humans and animals. Like most arboviruses, no medication is available to treat the disease caused by SLEV, let alone vaccines. Therefore, repellent and other mosquito containment measures are the best ways to prevent infection. Currently, inhibitory biological agents, such as the Wolbachia bacterium, are seen as potential tools for the biological control of arboviruses. In this context, ISVs have been identified as possible new biological agents to control arbovirus transmission. Our results show that BREV can suppress SLEV in *Aedes albopictus* mosquito cells; this preliminary information is important in the use of BREV as a vector control for SLEV in the future.

It is essential to continue this study by conducting in vivo experiments on mosquitoes to validate the results obtained in vitro and evaluate their possible application in public health.

## 5. Conclusions

The results of the present study show the ability of BREV to suppress SLEV replication in *Aedes albopictus* cells, clone C6/36 cells, with this interference being dose-dependent. SLEV replication did not interfere with BREV replication. Future studies in live mosquitoes are encouraged to validate the suppression caused by BREV in SLEV replication.

## Figures and Tables

**Figure 1 viruses-16-00210-f001:**
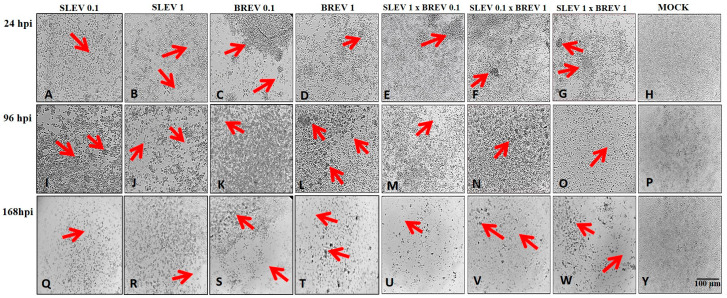
Micrographs of Aedes albopictus mosquito cells (C6/36) showing the CPE caused by the Saint Louis encephalitis virus (SLEV) and the Brejeira virus (BREV), carried out in the three combinations of MOIs used in the co-infection (SLEV 0.1 × BREV 1; SLEV 1 × BREV 0.1; SLEV 1 × BREV 1), as well as the respective positive controls for SLEV and BREV (in the different MOIs) and negative controls (Mock). Viral kinetics are from 0 to 168 h post-infection, but, here, the highlights are 24 hpi, 96 hpi, and 168 hpi. The red arrows highlight the characteristics of the CPE. (**A**,**B**,**J**,**K**,**M**,**O**,**Q**,**R**) Destruction of the cell monolayer; (**C**,**S**,**T**,**U**,**V**,**W**) clusters of cells in suspension and destruction of the cell monolayer; (**D**–**F**,**N**) formation of clumps of cells; (**H**,**P**,**Y**) absence of CPE; (**G**,**I**) clusters of cells adhered to the cell monolayer and destruction of the cell monolayer; and (**L**) clusters of cells in suspension.

**Figure 2 viruses-16-00210-f002:**
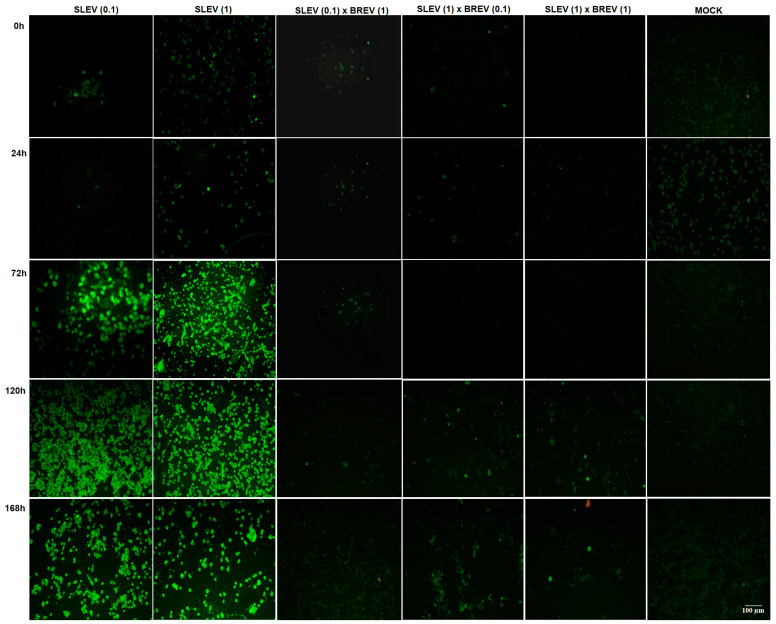
Micrographs of the detection of Saint Louis encephalitis virus (SLEV) by indirect immunofluorescence in the three combinations of MOIs used in co-infection with Brejeira virus (BREV) (SLEV 0.1 × BREV 1; SLEV 1 × BREV 0.1; SLEV 1 × BREV 1), as well as the respective positive and negative controls (Mock). Viral kinetics from 0 to 168 h post-infection, evidencing the 0 hpi, 24 hpi, 72 hpi, 120 hpi, and 168 hpi.

**Figure 3 viruses-16-00210-f003:**
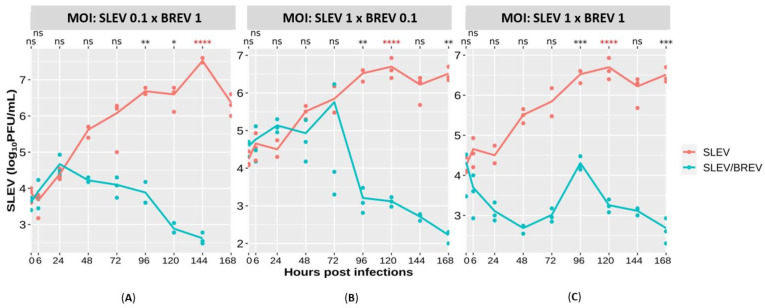
SLEV viral titers obtained by plaque assay on Vero cells, comparing the co-infections with their respective positive controls, using different concentrations of MOIs for both the arbovirus (SLEV) and BREV. (**A**) Viral titers with MOI of 0.1 for SLEV and MOI of 1 for BREV; (**B**) viral titers with MOI of 1 for SLEV and MOI of 0.1 for BREV; and (**C**) titers with MOI of 1 for both viruses. ns: not significant; * significant; ** very significant; *** and **** extremely significant.

**Figure 4 viruses-16-00210-f004:**
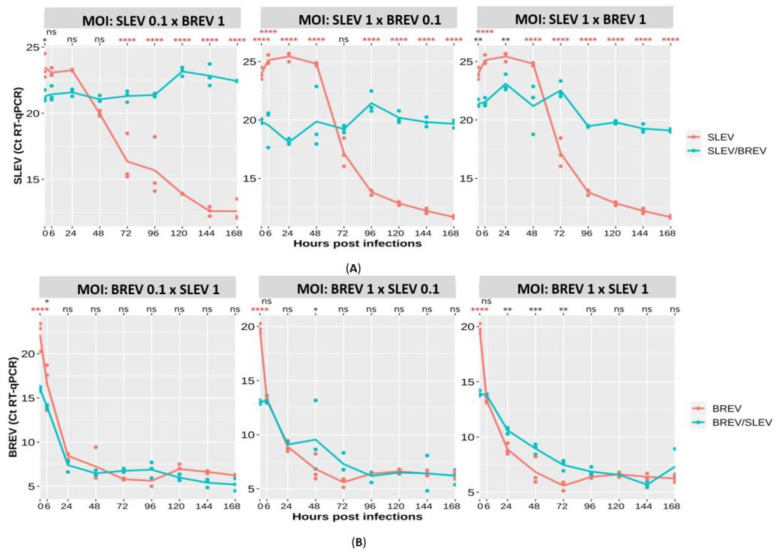
Detection of Ct by RT-qPCR for the SLEV and BREV viruses using MOI concentrations of 0.1 and 1, considering co-infections and positive controls of the respective viruses. (**A**) Ct by RT-qPCR for SLEV. (**B**) Ct by RT-qPCR for BREV. ns: not significant; * significant; ** very significant; *** and **** extremely significant.

**Table 1 viruses-16-00210-t001:** Primers and probes used in RT-PCR for SLEV and BREV.

Virus	Primers/Probes
SLEV	SLE2420 2420–2439 CTGGCTGTCGGAGGGATTCT 68 (Primer)
	SLE2487c 2487–2468 TAGGTCAATTGCACATCCCG (Primer)
	SLE2444-probe 2444–2466 TCTGGCGACCAGCGTGCAAGCCG (Probe)
	SLE834 834–852 GAAAACTGGGTTCTGCGCA 72 (Primer)
	SLE905c 905–889 GTTGCTGCCTAGCATCCATCC (Primer)
	SLE857-probe 857–880 TGGATATGCCCTAGTTGCGCTGGC (Probe)
BREV	BREV–F–ATGACCGATGATGAGAACCG (Primer)
	BREV–R–GGTGAGACAGCAATAGTAGCC (Primer)
	BREV–Probe–TCGTGCTCGATGACACCCGC (Probe)

## Data Availability

Data are contained within the article.

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
