# Peer review of "Viral Interference between the Insect-Specific Virus Brejeira and the Saint Louis Encephalitis Virus In Vitro"

_viruses, 2024, doi:10.3390/v16020210_

Round 1

Reviewer 1 Report

Comments and Suggestions for Authors

The authors describe a study that was conducted to determine if the insect specific Negevirus, Brejeira (BREV) interfered with St. Louis encephalitis (SLE) virus infection and /or that SLE virus interfered with BREV virus infection in Aedes albopictus C6/36 cells. The cells were co-infected using different multiplicity of infection up to 7 days for each of the following conditions: 1) Concurrent infection with BREV and SLEV; 2) Infection with BREV only (positive control); 3) Infection with SLEV only (positive control); 4) Negative Control. Subsequently, the cells were observed under an inverted optical microscope for possible cytopathic effect (CPE) in the cell monolayer, and cells were collected on slides for indirect immunofluorescence assay (IFA) immediately after inoculation; 06 hours; 24 hours; 48 195 hours; 72 hours, 96 hours, 120 hours, 144 hours and, 168 hours post-infection, and suspensions of infected cells were also collected at each time interval of the established viral kinetics. Cell suspensions collected in the kinetics were titrated by plaque by assay on Vero cells to compare the SLEV titer in the SLEV positive control and co-infection conditions. Also, RNA was extracted from the cell supernatant to obtain the number of viral RNA copies using the reverse transcription technique (RT-PCR). The authors claimed that (CPE) caused by BREV was more pronounced than the CPE caused by SLEV in C6/36 cells, and suppression of SLEV replication was also observed in cells co-infected with both viruses for all the MOIs used in the experiments, but the extent of the interference was dose-dependent. Therefore, the conclusion was that ISV BREV had the ability to suppress SLEV replication in Aedes albopictus cells and that SLEV did not interfere with BREV replication. Overall, the manuscript is readily understood and the contents, including the scientific merits are suitable for publication consideration in the Journal “Viruses”. The abstract only includes results that showed CPE caused by BREV to be more pronounced than the CPE caused by SLEV in C6/36 cells, and suppression of SLEV replication was also observed in the co-infection condition in all the MOIs used. The inclusion of a brief summary of the results for the IFI, plaque assay, and RT-PCR would improve the contents of the abstract. The contents of the Introduction adequately described the current understanding of the subject matter and describes previous studies that showed other ISV to interfere with the replication of other arboviruses. The methods are adequately described and appropriate for achieving the results of the study. The results are adequately described and present an excellent description of the observations with the exception of the quality of Figure 1 that is not readily understood. The discussion addresses the main observations in relation to other published studies but could be improved by removing much of the results that duplicates the contents of the results section. Overall, this manuscript is well prepared and contributes further to the understanding of the interaction between a Negev virus and the well studied SLEV. The authors recognize the critical need to conduct further studies in-vivo in mosquitoes in order to validate the results obtained in vitro and, consequently, evaluate their possible application in the context of public health.

Specific comments

Lines 78-80, The authors state that “The first isolate of SLEV was obtained from a brain suspension of a case of acute 78 encephalitis during a large urban outbreak of the disease in the city of Saint Louis, United 79 States, in 1933” (Reference needed here).

Lines 89-90, The authors state that “The SLEV maintenance cycle involves mosquitoes of the Culex genus, mainly Culex declarator and Culex coronator,….The authors should consider adding  Culex pipiens, Culex quinquefasciatus and Culex tarsalis in North America.

Lines 127-128, The authors refer to the reference #16 as indicating that ISV can be used to ISV can be used as a positive tool in the fight against these major public health problems. This is misleading because the contents of the article does not conduct any relevant studies regarding the use of ISV to fight against health problems associated with arboviral infections. The contents of the discussion did however address observations made on the prevention of dengue, Chikungunya and West Nile virus infection by Wolbachia infected Aedes aegypti, but again nothing about the use of ISV to prevent and/or control arboviral diseases..

Lines 528 and 529 reads, “ESVL has 528 emergency potential in Brazil and worldwide, and is responsible for causing encephalitogenic diseases in humans and animals”. The reviewer assume this is a typo and should read SLEV rather than ESLV.

Line 161, The authors state, “from Culex (Cux.) mosquitoes”, this should read (Cx.) as the abbreviation for Culex.

Lines 459-464, The authors states that “BREV showed a well-defined CPE with the formation of suspended clumps, elliptical cells, and loose dead cells, which had already been reported by Ribeiro and colleagues (2022) [23]. Our experiments showed that the characteristic CPE of BREV stood out over the CPE of SLEV, which demonstrates the interference caused by BREV in SLEV replication, regardless of the concentrations of MOIs used. On the other hand, the replication of negevirus shows no interference when placed in the presence of SLEV in the same condition (Figure 1)

In reference to lines 459-464, The reviewer does not understand why the 2022 before reference #23 and suggest that the past tense  be used for, “which demonstrated….” and past tense for “On the other hand, the replication of negevirus showed no interference.

Comments on the Quality of English Language

Overall, good.

Author Response

All the alterations in the main text are highlighted in yellow.

Reviewer 1:

English: Minor editing of English language required.

Answer: A native English speaker revised the manuscript.

The abstract only includes results that showed CPE caused by BREV to be more pronounced than the CPE caused by SLEV in C6/36 cells, and suppression of SLEV replication was also observed in the co-infection condition in all the MOIs used. The inclusion of a brief summary of the results for the IFI, plaque assay, and RT-PCR would improve the contents of the abstract.

Answer: Thanks for the suggestion. We missed this information in our abstract. We included this.

The results are adequately described and present an excellent description of the observations with the exception of the quality of Figure 1 that is not readily understood.

Answer: Thanks for your recommendation. We included this information in the title of Figure 1.

The discussion addresses the main observations in relation to other published studies but could be improved by removing much of the results that duplicates the contents of the results section.

Answer: Thanks for your recommendation. We deleted part of the description of the studies results from the discussion section.

Specific comments

Lines 78-80, The authors state that “The first isolate of SLEV was obtained from a brain suspension of a case of acute 78 encephalitis during a large urban outbreak of the disease in the city of Saint Louis, United 79 States, in 1933” (Reference needed here).

Answer: We added the reference at the end of this paragraph.

 Lines 89-90, The authors state that “The SLEV maintenance cycle involves mosquitoes of the Culex genus, mainly Culex declarator and Culex coronator,….The authors should consider adding  Culex pipiens, Culex quinquefasciatus and Culex tarsalis in North America.

 Answer: We included the information in this paragraph.

Lines 127-128, The authors refer to the reference #16 as indicating that ISV can be used to ISV can be used as a positive tool in the fight against these major public health problems. This is misleading because the contents of the article does not conduct any relevant studies regarding the use of ISV to fight against health problems associated with arboviral infections. The contents of the discussion did however address observations made on the prevention of dengue, Chikungunya and West Nile virus infection by Wolbachia infected Aedes aegypti, but again nothing about the use of ISV to prevent and/or control arboviral diseases..

 Answer: This is right. We corrected this. We chose another reference to this paragraph, reference 17 (Vasilakis and Tesh, 2015).

Lines 528 and 529 reads, “ESVL has 528 emergency potential in Brazil and worldwide, and is responsible for causing encephalitogenic diseases in humans and animals”. The reviewer assume this is a typo and should read SLEV rather than ESLV.

 Answer: Thanks for the observation. You are right. We corrected it.

Line 161, The authors state, “from Culex (Cux.) mosquitoes”, this should read (Cx.) as the abbreviation for Culex.

 Answer: Thanks for the observation. You are right. We corrected it.

Lines 459-464, The authors states that “BREV showed a well-defined CPE with the formation of suspended clumps, elliptical cells, and loose dead cells, which had already been reported by Ribeiro and colleagues (2022) [23]. Our experiments showed that the characteristic CPE of BREV stood out over the CPE of SLEV, which demonstrates the interference caused by BREV in SLEV replication, regardless of the concentrations of MOIs used. On the other hand, the replication of negevirus shows no interference when placed in the presence of SLEV in the same condition (Figure 1). In reference to lines 459-464, The reviewer does not understand why the 2022 before reference #23 and suggest that the past tense  be used for, “which demonstrated….” and past tense for “On the other hand, the replication of negevirus showed no interference.

Answer: Thank you. We corrected this.

Reviewer 2 Report

Comments and Suggestions for Authors

In this manuscript the authors presume to demonstrate the ability of an insect specific virus (Brejeira virus of the negevirus family) to interfere with the arbovirus, Saint Louis encephalitis virus of the Flaviviridae family, which can cause significant human disease. Both of these viruses are most commonly associated with Culex species mosquitoes; however, the authors chose to use a cloned C6/36 cell line which is derived from Aedes albopictus mosquitoes for their studies.

The authors clearly demonstrate virus induced cytophathic effects in C6/36 cells, which is troublesome since arthropod-borne viruses typically do not kill their host and significantly confounds their results. More specifically, the authors clearly demonstrate that BREV virus infection alone results in more significant CPE within the first 24 to 48 hours of infection and peak viral titers between 48 and 72 hours regardless of the initial MOI. These kinetics are not altered significantly during co-infection. In contrast, infection of C6/36 cells with SLEV alone results in delayed CPE with peak viral titers between 144 to 168 hours; and these kinetics are significantly altered during co-infection with BREV such that SLEV does not appear to replicate at all. The problem is, if the cells are dead witin 24 to 48 hours of infection with the more virulent BREV then the observed decrease in SLEV is as likely due to a lack of viable cells capable of supporting SLEV replication as opposed to direct viral intereference. The authors do not discuss this possibility in the results or discussion. This should also be taken into cosideration when discussing results from previous studies demonstrating ISV intereference with arbovirus infections when the viruses were either from the same family and/or have the same viral replication kinetics.

Other obersvations: An MOI of 0.1 should be used throughout as opposed to 0,1.

Line 72: an arbovirus is simply an arthropod-borne virus it is not the disease. 

Line 76: the viral particle is spherical and possess an icosahedral nucleocapsid.

Line 87: antibodies can be detected in 1 to 5% of what animal? Humans?

Lines 120 - 124: run on sentence. rewrite.

Line 125: these are viruses that cause outbreaks, not diseases.

Line 164: Cloned, not clone.

2.3 Preparation of viral stocks and titration: How were the stocks characterized to confirm identity and purity? What was the TCID50 method used for. There is no mention of TCID50 titers in the manuscript.

2.5 co-infection experiments: How were infections carried out? What volume was used for infection and for what period of time? Was the inoculum removed and the cells washed before adding media? How was the supernatant collected? Was it removed from a well and the media replaced or were the cells and inoculum collected from different wells for each time point?

Line 217: reverse transcription technique followed by PCR. (The RT is redundant).

Line 232: I'm not familiar with a Way ANOVA. This should be one-way or two-way?

Line 237: If BREV doesn't produce CPE, how was the PFU titer determined? Was this actuall TCID50/ml titer?

Figure 1: what are the red arrows showing?

Figure 2: what is ESLV? Should this be SLEV? ESLV is also used elsewhere in the manuscript itself.

Line 367: real-time is redundant if using the RT-PCR abbreviation.

Discussions and conclusions need to be rewritten in the context of the fact that C6/36 cells die as a result of BREV infection long before SLEV has a chance to replicate.

Comments on the Quality of English Language

While readable, the manuscript would benefit from significant editing for grammar to improve comprehension.

Author Response

All the alterations in the main text are highlighted in yellow.

 Reviewer 2:

English: Extensive editing of English language required.

Answer: A native English speaker revised the manuscript.

Both of these viruses are most commonly associated with Culex species mosquitoes; however, the authors chose to use a cloned C6/36 cell line which is derived from Aedes albopictus mosquitoes for their studies.

Answer: Culex species mosquitoes are the primary vector of both SLEV and BREV. We used C6/36 cells because it is a well-established culture for growing negeviruses such as the BREV, reaching high titers in this culture, as mentioned by Vasilakis and colleagues (2013). The C6/36 cells are also an excellent culture widely used for arbovirus diagnosis and studies; this applies to SLEV. Both virus strains in our study replicate very well in C6/36 cells. Our aim for the following studies is to coinfect Culex mosquitoes (in vivo).   

The authors clearly demonstrate virus induced cytophathic effects in C6/36 cells, which is troublesome since arthropod-borne viruses typically do not kill their host and significantly confounds their results. The problem is, if the cells are dead witin 24 to 48 hours of infection with the more virulent BREV then the observed decrease in SLEV is as likely due to a lack of viable cells capable of supporting SLEV replication as opposed to direct viral intereference. The authors do not discuss this possibility in the results or discussion. This should also be taken into cosideration when discussing results from previous studies demonstrating ISV intereference with arbovirus infections when the viruses were either from the same family and/or have the same viral replication kinetics.

Answer: It is true that most of arthropod-borne viruses commonly do not kill their host, and we observe this in the laboratory when the majority of arboviruses do not cause CPE in C6/36 cells, despite they replicate very well in these cells while others as Dengue virus, cause non-lytic effect in those cells, for example the formation of syncytia. However, insect-specific viruses are seen as possible ancestors of arboviruses, gaining the ability to infect humans. Then, the ISVs are better adapted to mosquito cells than arboviruses, especially the negeviruses, which, according to Vasilakis and colleagues (2013) and Nunes and colleagues (2017), cause lytic CPE in C6/36 cells.

Regarding the possibility of the cells dying within 24 to 48 hours of infection with BREV, we inoculated both viruses at the same time, concomitantly, trying to give them equal opportunity to infect the cells, using equal MOI between both viruses as well as different MOI combinations. The BREV kills part of the cells because it has a competitive advantage compared to SLEV. Observing the cells at the microscope daily, we saw that part of the cells were dead and flowing, while another part was alive and adhered to the well (you can see in the pictures of Figure 1, 96 hpi time point). The main aim of our study is to verify if the insect-specific virus interferes with the arbovirus replication, and our study showed that; however, our study is preliminary, and further studies in vivo using live mosquitoes are needed to validate these results. In the conclusion, we added the need for further studies in mosquitoes to validate our data.

Other observations: An MOI of 0.1 should be used throughout as opposed to 0,1.

Answer: Thanks for your observation. It's true. We corrected this in the figures.

Line 72: an arbovirus is simply an arthropod-borne virus it is not the disease. 

Answer: We corrected this.

Line 76: the viral particle is spherical and possess an icosahedral nucleocapsid.

Answer: Thanks. We corrected this.

Line 87: antibodies can be detected in 1 to 5% of what animal? Humans?

Answer: Humans. We clarified this sentence in the manuscript.

Lines 120 - 124: run on sentence. rewrite.

Answer: We rewrote this paragraph.

Line 125: these are viruses that cause outbreaks, not diseases.

Answer: Thanks for your observation. Its true. We corrected this.

Line 164: Cloned, not clone.

Answer: Thanks for your recommendation, but we disagree. The name of the cell culture is clone C6/36. Trying to improve we add the article “The” in the front of clone C6/36”. We hope this works.

2.3 Preparation of viral stocks and titration: How were the stocks characterized to confirm identity and purity? What was the TCID50 method used for. There is no mention of TCID50 titers in the manuscript.

Answer: Both viruses’ strains used in the study were sequenced and concentrated by ultracentrifugation. We used TCID50 to obtain the Brejeira virus titer because this virus does not replicate in Vero cells so we couldn’t use Plaque assay. The Brejeira virus causes a prominent cytopathic effect (CPE) in C6/36 cells, so we used TCID50 to obtain the virus titer by observing the CPE. We included the TCID50 method in the manuscript.

2.5 co-infection experiments: How were infections carried out? What volume was used for infection and for what period of time? Was the inoculum removed and the cells washed before adding media? How was the supernatant collected? Was it removed from a well and the media replaced or were the cells and inoculum collected from different wells for each time point?

Answer: One day after we seeded C6/36 cells in 12-well plates, the cells were infected concomitantly with SLEV and BREV (triplicate). Before the infection, we discarded the plates' growth media and inoculated the viruses. The volume inoculated in each well was defined depending on the MOI used in that well (we used the MOI formula to find the volume of virus to be inoculated in the wells according to the number of cells counted). After the inoculation, the cells were incubated for one hour in the incubator at 28ºC, and then we added maintenance medium to the plates. Afterward, we incubated the plates for 7 days and collected three wells of the coinfected cells per day. In our experiments, we also included positive and negative controls (in triplicate) for both viruses. The inoculum was not removed, and the cells were not washed before adding media. We collected not only the supernatant; the cells were scrapped, and the cell suspension was collected. The cell suspension (cells + supernatant) of each time point was collected from different wells. The media was not replaced. We included more information in the text.

Line 217: reverse transcription technique followed by PCR. (The RT is redundant).

Answer: Thanks for the observation. We corrected this.

Line 232: I'm not familiar with a Way ANOVA. This should be one-way or two-way?

Answer: It is two-way ANOVA. It is wrong in the manuscript. We included the information.

Line 237: If BREV doesn't produce CPE, how was the PFU titer determined? Was this actuall TCID50/ml titer?

Answer: We transformed the BREV titer obtained in TCID50/ml to PFU/ml using the formula: TCID50 titer (per mL) by 0.7 to predict the mean number of PFU/ml. This formula is available in Virology Culture Guide from ATCC (https://www.atcc.org/resources/culture-guides/virology-culture-guide#:~:text=For%20any%20titer%20expressed%20as,%C3%97%20105%20PFU%2FmL.). We included this information in the manuscript.

Figure 1: what are the red arrows showing?

Answer: The red arrows show the CPE caused by the viruses. We included the CPE of each image in the figure title. Thanks for your observation.

Figure 2: what is ESLV? Should this be SLEV? ESLV is also used elsewhere in the manuscript itself.

Answer: The correct is SLEV. The ESLV is wrong. We corrected this in the manuscript. 

Line 367: real-time is redundant if using the RT-PCR abbreviation.

Answer: Thanks for your observation. We corrected this.

Discussions and conclusions need to be rewritten in the context of the fact that C6/36 cells die as a result of BREV infection long before SLEV has a chance to replicate.

Answer: Regarding the possibility of the cells dying within 24 to 48 hours of infection with BREV, we inoculated both viruses at the same time, concomitantly, trying to give them equal opportunity to infect the cells, using equal MOI between both viruses as well as different MOI combinations. The BREV kills part of the cells because it has a competitive advantage compared to SLEV. Observing the cells at the microscope daily, we saw that part of the cells were dead and flowing, while another part was alive and adhered to the well (you can see in the pictures of Figure 1, 96 hpi time point). The main aim of our study is to verify if the insect-specific virus interferes with the arbovirus replication, and our study showed that; however, our study is preliminary, and further studies in vivo using live mosquitoes are needed to validate these results. In the conclusion, we added the need for further studies in mosquitoes to validate our data.

Round 2

Reviewer 2 Report

Comments and Suggestions for Authors

The reviewers concerns have been addressed adequately.

Comments on the Quality of English Language

Much improved.